# DMV3D: Denoising Multi-View Diffusion using 3D Large Reconstruction Model

**Yinghao Xu**[1,2*]  **Hao Tan**[1]  **Fujun Luan**[1]  **Sai Bi**[1]  **Peng Wang**[1,3]  **Jiahao Li**[1,5]
**Zifan Shi**[1,4]  **Kalyan Sunkavalli**[1]  **Gordon Wetzstein**[2]  **Zexiang Xu**[1†]  **Kai Zhang**[1†]
[1]Adobe Research  [2]Stanford  [3]HKU  [4]HKUST  [5]TTIC

## ABSTRACT

We propose *DMV3D*, a novel 3D generation approach that uses a transformer-based 3D large reconstruction model to denoise multi-view diffusion. Our reconstruction model incorporates a triplane NeRF representation and can denoise noisy multi-view images via NeRF reconstruction and rendering, achieving single-stage 3D generation in $\sim$30s on single A100 GPU. We train *DMV3D* on large-scale multi-view image datasets of highly diverse objects using only image reconstruction losses, without accessing 3D assets. We demonstrate state-of-the-art results for the single-image reconstruction problem where probabilistic modeling of unseen object parts is required for generating diverse reconstructions with sharp textures. We also show high-quality text-to-3D generation results outperforming previous 3D diffusion models. Our project website is at: https://justimyhxu.github.io/projects/dmv3d/.

## 1 INTRODUCTION

The advancements in 2D diffusion models (Ho et al., 2020; Song et al., 2020a; Rombach et al., 2022a) have greatly simplified the image content creation process and revolutionized 2D design workflows. Recently, diffusion models have also been extended to 3D asset creation in order to reduce the manual workload involved for applications like VR, AR, robotics, and gaming. In particular, many works have explored using pre-trained 2D diffusion models for generating NeRFs (Mildenhall et al., 2020) with score distillation sampling (SDS) loss (Poole et al., 2022; Lin et al., 2023a). However, SDS-based methods require long (often hours of) per-asset optimization and can frequently lead to geometry artifacts, such as the multi-face Janus problem.

On the other hand, attempts to train 3D diffusion models have also been made to enable diverse 3D asset generation without time-consuming per-asset optimization (Nichol et al., 2022; Jun & Nichol, 2023). These methods typically require access to ground-truth 3D models/point clouds for training, which are hard to obtain for real images. Besides, the latent 3D diffusion approach (Jun & Nichol, 2023) often leads to an unclean and hard-to-denoise latent space (Chen et al., 2023b) on highly diverse category-free 3D datasets due to two-stage training, making high-quality rendering a challenge. To circumvent this, single-stage models have been proposed (Anciukevičius et al., 2023; Karnewar et al., 2023), but are mostly category-specific and focus on simple classes.

Our goal is to achieve fast, realistic, and generic 3D generation. To this end, we propose DMV3D, a novel single-stage category-agnostic diffusion model that can generate 3D (triplane) NeRFs from text or single-image input conditions via direct model inference. Our model allows for the generation of diverse high-fidelity 3D objects within 30 seconds per asset (see Fig. 1). In particular, DMV3D is a 2D multi-view image diffusion model that integrates 3D NeRF reconstruction and rendering into its denoiser, trained without direct 3D supervision, in an end-to-end manner. This avoids both separately training 3D NeRF encoders for latent-space diffusion (as in two-stage models) and tedious per-asset optimization (as in SDS methods).

In essence, our approach uses a 3D reconstruction model as the 2D multi-view denoiser in a multi-view diffusion framework. This is inspired by RenderDiffusion (Anciukevičius et al., 2023) –

---

[*]This work is done while the author is an intern at Adobe Research.

[†]denotes equal advisory.

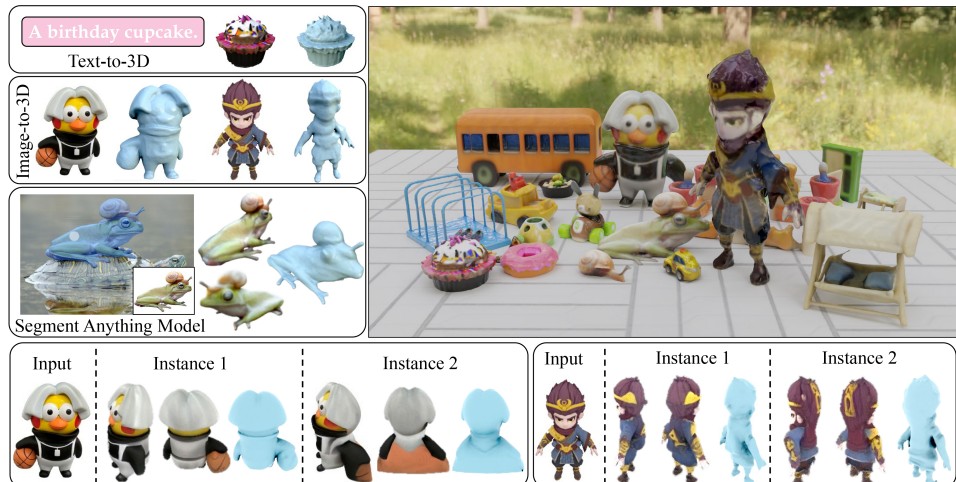

Figure 1: Top left: our approach achieves fast 3D generation (∼30s on A100 GPU) from text or single-image input; the latter one, combined with 2D segmentation methods (like SAM (Kirillov et al., 2023)), can reconstruct objects segmented from natural images. Bottom: as a probabilistic single-image-to-3D model, we can produce multiple reasonable 3D assets from the same image. Top right: we demonstrate a scene comprising diverse 3D objects generated by our models.

achieving 3D generation through single-view diffusion. However, their single-view framework relies on category-specific priors and canonical poses and thus cannot easily be scaled up to generate arbitrary objects. In contrast, we consider a sparse set of four multi-view images that surround an object, adequately describing a 3D object without strong self-occlusions. This design choice is inspired by the observation that humans can easily imagine a complete 3D object from a few surrounding views with little uncertainty. However, utilizing such inputs essentially requires addressing the task of sparse-view 3D reconstruction – a long-standing problem and known to be highly challenging even without noise in the inputs.

We address this by leveraging large transformer models that have been shown to be effective and scalable in solving various challenging problems (Jun & Nichol, 2023; Nichol et al., 2022; Hong et al., 2023; Brown et al., 2020; Shen et al., 2023). In particular, built upon the recent 3D Large Reconstruction Model (LRM) (Hong et al., 2023), we introduce a novel model for joint reconstruction and denoising. More specifically, our transformer model can, from a sparse set of noisy multi-view images, reconstruct a clean (noise-free) NeRF model that allows for rendering (denoised) images at arbitrary viewpoints. Our model is conditioned on the diffusion time step, designed to handle any noise levels in the diffusion process. It can thus be directly plugged as the multi-view image denoiser in an multi-view image diffusion framework.

We enable 3D generation conditioned on single images/texts. For image conditioning, we fix one of the sparse views as the noise-free input and denoise other views, similar to 2D image inpainting (Xie et al., 2023). We apply attention-based text conditioning and classifier-free guidance, commonly used in 2D diffusion models, to enable text-to-3D generation. We train our model on large-scale datasets consisting of both synthetic renderings from Objaverse (Deitke et al., 2023) and real captures from MVImgNet (Yu et al., 2023) with only image-space supervision. Our model achieves state-of-the-art results on single-image 3D reconstruction, outperforming prior SDS-based methods and 3D diffusion models. We also demonstrate high-quality text-to-3D results outperforming previous 3D diffusion models. In sum, our main contributions are:

- A novel single-stage diffusion framework that leverages multi-view 2D image diffusion model to achieve 3D generation;
- An LRM-based multi-view denoiser that can reconstruct noise-free triplane NeRFs from noisy multi-view images;
- A general probabilistic approach for high-quality text-to-3D generation and single-image reconstruction that uses fast direct model inference (∼30s on single A100 GPU).

Our work offers a novel perspective to address 3D generation tasks, which bridges 2D and 3D generative models and unifies 3D reconstruction and generation. This opens up opportunities to build a foundation model for tackling a variety of 3D vision and graphics problems.

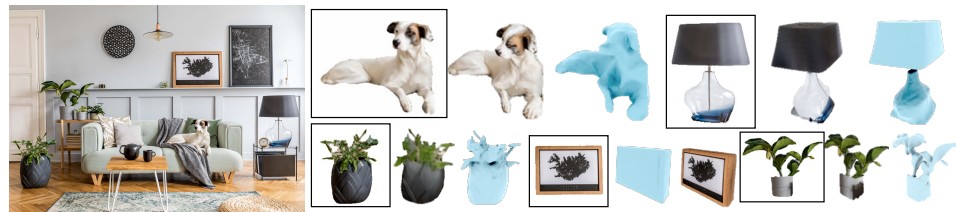

Figure 2: **SAM + DMV3D.** We can use SAM (Kirillov et al., 2023) to segment any objects from a real scene photo and reconstruct their 3D shape and appearance with our method, showcasing our model's potential in enabling 3D-aware image editing experiences.

## 2 RELATED WORK

**Sparse-view Reconstruction.** Neural representations (Mescheder et al., 2019; Park et al., 2019; Mildenhall et al., 2020; Sitzmann et al., 2019; 2020; Chen et al., 2022; Müller et al., 2022) offer a promising platform for scene representation and neural rendering (Tewari et al., 2022). Applied to novel-view synthesis, these approaches have been successful in single-scene overfitting scenarios where lots of multi-view training images are available. Recent efforts (Yu et al., 2021; Chen et al., 2021; Long et al., 2022; Wang et al., 2021; Lin et al., 2023b; Jain et al., 2021) have extended these ideas to operate with a sparse set of views, showcasing improved generalization capabilities to unseen scenes. As non-generative methods, these approaches struggle on covering the multiple modes in the large-scale datasets and thus can not generate diverse realistic results. In particular, the recently-proposed LRM (Hong et al., 2023) tackles the inherent ambiguous single-image-to-3D problem in a deterministic way, resulting in blurry and washed-out textures for unseen part of the objects due to mode averaging. We resolve this issue by building a probabilistic image-conditioned 3D generation model through denosing multi-view diffusion.

**3D Generative Adversarial Networks (GANs).** GANs have made remarkable advancements in 2D image synthesis (Brock et al., 2018; Karras et al., 2018; 2019; 2020; 2021). 3D GANs (Nguyen-Phuoc et al., 2019; Schwarz et al., 2020; Chan et al., 2021; 2022; Niemeyer & Geiger, 2021; Gu et al., 2021; Skorokhodov et al., 2022; Xu et al., 2022; 2023; Shi et al., 2022; Gao et al., 2022; Skorokhodov et al., 2023) extend these capabilities to generating 3D-aware assets from unstructured collections of single-view 2D images in an unsupervised manner. GAN architectures, however, are difficult to train and generally best suited for modeling datasets of limited scale and diversity (Dhariwal & Nichol, 2021).

**3D-aware Diffusion Models (DMs).** DMs have emerged as foundation models for visual computing, offering unprecedented quality, fine-grained control, and versatility for 2D image generation (Ho et al., 2020; Song et al., 2020b; Rombach et al., 2022a; Po et al., 2023). Several strategies have been proposed to extend DMs to the 3D domain. Some of these approaches (Nichol et al., 2022; Jun & Nichol, 2023; Shue et al., 2023; Gupta et al., 2023; Ntavelis et al., 2023) use direct 3D supervision. The quality and diversity of their results, however, is far from that achieved by 2D DMs. This is partly due to the computational challenge of scaling diffusion network models up from 2D to 3D, but perhaps more so by the limited amount of available 3D training data. Other approaches in this category build on optimization using a differentiable 3D scene representation along with the priors encoded in 2D DMs (Poole et al., 2022; Lin et al., 2023a; Wang et al., 2022; 2023). While showing some success, the quality and diversity of their results is limited by the SDS–based loss function (Poole et al., 2022). Another class of methods uses 2D DM–based image-to-image translation using view conditioning (Liu et al., 2023b; Chan et al., 2023; Gu et al., 2023). While these approaches promote multi-view consistency, they do not enforce it, leading to flicker and other view-inconsistent effects. Finally, several recent works have shown success in training 3D diffusion models directly on single-view or multi-view image datasets (Karnewar et al., 2023; Chen et al., 2023b; Shen et al., 2023) for relatively simple scenes with limited diversity.

Prior RenderDiffusion (Anciukevičius et al., 2023) and concurrent Viewset Diffusion (Szymanowicz et al., 2023) work are closest to our method. Both solve the 3D generation problem using 2D DMs with 3D-aware denoisers. Neither of these methods, however, has been demonstrated to work on highly diverse datasets containing multi-view data of $> 1M$ objects. Our novel LRM-based (Hong et al., 2023) 3D denoiser architecture overcomes this challenge and enables state-of-the-art results for scalable, diverse, and high-quality 3D generation.

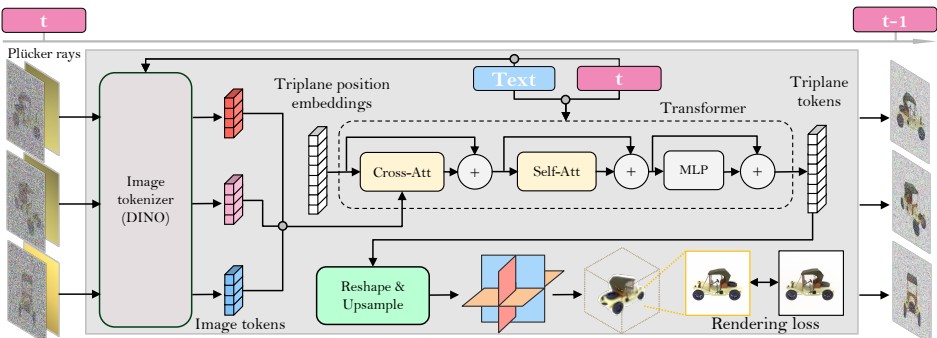

Figure 3: **Overview of our method.** We denoise multiple views (three shown in the figure to reduce clutterness; four used in experiments) for 3D generation. Our multi-view denoiser is a large transformer model that reconstructs a noise-free triplane NeRF from input noisy images with camera poses (parameterized by Plucker rays). During training, we supervise the triplane NeRF with a rendering loss at input and novel viewpoints. During inference, we render denoised images at input viewpoints and combine them with inputs to obtain less noisy inputs for the next denoising step. We output the clean triplane NeRF at final denoising step, enabling 3D generation. Refer to Sec. 3.3 for how to extend this model to condition on single image or text.

# 3 METHOD

We now present our single-stage 3D diffusion model. In particular, we introduce a novel diffusion framework that uses a reconstruction-based denoiser to denoise noisy multi-view images for 3D generation (Sec. 3.1). Based on this, we propose a novel LRM-based (Hong et al., 2023) multi-view denoiser conditioning on diffusion time step to progressively denoise multi-view images via 3D NeRF reconstruction and rendering (Sec. 3.2). We further extend our model to support text and image conditioning, enabling controllable generation (Sec. 3.3).

## 3.1 MULTI-VIEW DIFFUSION AND DENOISING

**Diffusion.** Denoising Diffusion Probabilistic Models (DDPM) transforms the data distribution $x_0 \sim q(x)$ using a Gaussian noise schedule in the forward diffusion process. The generation process is the reverse process where images are gradually denoised. The diffused data sample $x_t$ at timestep $t$ can be written as $x_t = \sqrt{\bar{\alpha}_t} x_0 + \sqrt{1 - \bar{\alpha}_t} \epsilon$, where $\epsilon \sim \mathcal{N}(0, \mathbf{I})$ represents Gaussian noise and the monotonically decreasing $\bar{\alpha}_t$ controls the Signal-Noise-Ratio (SNR) of noisy sample $x_t$.

**Multi-view diffusion.** The original $x_0$ distribution addressed in 2D DMs is the (single) image distribution in a dataset. We instead consider the (joint) distribution of multi-view images $\mathcal{I} = \{\mathbf{I}_1, ..., \mathbf{I}_N\}$, where each set of $\mathcal{I}$ are image observations of the same 3D scene (asset) from viewpoints $\mathcal{C} = \{c_1, ..., c_N\}$. The diffusion process is equivalent to diffusing each image independently with the same noise schedule:

$$\mathcal{I}_t = \{\sqrt{\bar{\alpha}_t} \mathbf{I} + \sqrt{1 - \bar{\alpha}_t} \epsilon_{\mathbf{I}} | \mathbf{I} \in \mathcal{I}\} \tag{1}$$

Note that this diffusion process is identical to the original one in DDPM, despite that we consider a specific type of data distribution $x = \mathcal{I}$ denoting per-object 2D multi-view images.

**Reconstruction-based denoising.** The reverse of the 2D diffusion process is essentially denoising. In this work, we propose to leverage 3D reconstruction and rendering to achieve 2D multi-view image denoising, while outputting a clean 3D model for 3D generation. In particular, we leverage a 3D reconstruction module $\mathrm{E}(\cdot)$ to reconstruct a 3D representation S from the noisy multi-view images $\mathcal{I}_t$, and render denoised images with a differentiable rendering module $\mathrm{R}(\cdot)$:

$$\mathbf{I}_{r,t} = \mathrm{R}(\mathrm{S}_t, c), \quad \mathrm{S}_t = \mathrm{E}(\mathcal{I}_t, t, \mathcal{C}) \tag{2}$$

where $\mathbf{I}_{r,t}$ represents a rendered image from $\mathrm{S}_t$ at a specific viewpoint $c$.

Denoising the multi-view input $\mathcal{I}_t$ is done by rendering $\mathrm{S}_t$ at the viewpoints $\mathcal{C}$, leading to the prediction of noise-free $\mathcal{I}_0$. This is equivalent to $x_0$ prediction in 2D DMs (Song et al., 2020a);

one can solve for $x_{t-1}$ from the input $x_t$ and prediction $x_0$ to enable progressive denoising during inference. However, unlike pure 2D generation, we find only supervising $\mathcal{I}_0$ prediction at input viewpoints cannot guarantee high-quality 3D generation (see Tab. 3), often leading to degenerate 3D solutions where input images are pasted on view-aligned planes. Therefore, we propose to supervise novel-view renderings from the 3D model $S_t$ as well, which leads to the following training objective:

$$L_{recon}(t) = \mathbb{E}_{\mathbf{I}, \boldsymbol{c} \sim \mathcal{I}_{full}, \mathcal{C}_{full}} \; \ell\big(\mathbf{I}, R(E(\mathcal{I}_t, t, \mathcal{C}), \boldsymbol{c})\big) \tag{3}$$

where $\mathcal{I}_{full}$ and $\mathcal{C}_{full}$ represent the full set of images and poses (from both randomly selected input and novel views), and $\ell(\cdot, \cdot)$ is an image reconstruction loss penalizing the difference between groundtruth $\mathbf{I}$ and rendering $R(E(\mathcal{I}_t, t, \mathcal{C}), \boldsymbol{c})$. Note that our framework is general – potentially any 3D representations (S) can be applied. In this work, we consider a (triplane) NeRF (Chan et al., 2022) representation (where $R(\cdot)$ becomes neural volumetric rendering (Mildenhall et al., 2020)) and propose a LRM-based reconstructor $E(\cdot)$ (Hong et al., 2023).

## 3.2 RECONSTRUCTOR-BASED MULTI-VIEW DENOISER

We build our multi-view denoiser upon LRM (Hong et al., 2023) and uses large transformer model to reconstruct a clean triplane NeRF (Chan et al., 2022) from noisy sparse-view posed images. Renderings from the reconstructed triplane NeRF are then used as denoising outputs.

**Reconstruction and Rendering.** As shown in Fig. 3, we use a Vision Transformer (DINO (Caron et al., 2021)) to convert input images $\mathcal{I} = \{\mathbf{I}_1, ..., \mathbf{I}_N\}$ to 2D tokens, and then use a transformer to map a learnable triplane positional embedding to the final triplane representing the 3D shape and appearance of an asset; the predicted triplane is then used to decode volume density and color with an MLP (not shown in Fig. 3 to avoid clutterness) for differentiable volume rendering. The transformer model consists of a series of triplane-to-images cross-attention and triplane-to-triplane self-attention layers as in the LRM work (Hong et al., 2023). We further enable time conditioning for diffusion-based progressive denoising and introduce a new technique for camera conditioning.

**Time Conditioning**. Our transformer-based model requires different designs for time-conditioning, compared to CNN-based DDPM (Ho et al., 2020). Inspired by DiT (Peebles & Xie, 2022), we condition on time by injecting the *adaLN-Zero block* (Ho et al., 2020) into the self- and cross-attention layers of our model to effectively handle inputs with different noise levels.

**Camera Conditioning**. Training our model on datasets with highly diverse camera intrinsics and extrinsics, e.g., MVImgNet (Yu et al., 2023), requires an effective design of input camera conditioning to facilitate the model's understanding of cameras for 3D reasoning. A basic strategy is, as in the case of time conditioning, to use *adaLN-Zero block* (Peebles & Xie, 2022) on the camera parameters (as done in Hong et al. (2023); Li et al. (2023)). However, we find that conditioning on camera and time simultaneously with the same strategy tends to weaken the effects of these two conditions and often leads to an unstable training process and slow convergence. Instead, we propose a novel approach – parameterizing cameras with sets of pixel-aligned rays. In particular, following Sitzmann et al. (2021); Chen et al. (2023a), we parameterize rays using Plucker coordinates as $\boldsymbol{r} = (\boldsymbol{o} \times \boldsymbol{d}, \boldsymbol{d})$, where $\boldsymbol{o}$ and $\boldsymbol{d}$ are the origin and direction of a pixel ray computed from the camera parameters, and $\times$ denotes cross-product. We concatenate the Plucker coordinates with image pixels, and send them to the ViT transformer for 2D image tokenization, achieving effective camera conditioning.

## 3.3 CONDITIONING ON SINGLE IMAGE OR TEXT

The methods described thus far enable our model to function as an unconditional generative model. We now introduce how to model the conditional probabilistic distribution with a conditional denoiser $E(\mathcal{I}_t, t, \mathcal{C}, y)$, where $y$ is text or image, enabling controllable 3D generation.

**Image Conditioning**. We propose a simple but effective image-conditioning strategy that requires no changes to our model architecture. We keep the first view $\mathbf{I}_1$ (in the denoiser input) noise-free to serve as the conditioning image, while applying diffusion and denoising on other views. In this case, the denoiser essentially learns to fill in the missing pixels within the noisy unseen views using cues extracted from the first input view, similar to the task of image inpainting which has been shown to be addressable by 2D DMs (Rombach et al., 2022a). In addition, to improve the generalizability of our

image-conditioned model, we generate triplanes in a coordinate frame aligned with the conditioning view and render other images using poses relative to the conditioning one. We normalize the input view's pose in the same way as LRM (Hong et al., 2023) during training, and specify the input view's pose in the same way too during inference.

**Text Conditioning**. To add text conditioning into our model, we adopt a strategy similar to that presented in Stable Diffusion (Rombach et al., 2022a). We use the CLIP text encoder (Radford et al., 2021) to generate text embeddings and inject them into our denoiser using cross-attention. Specifically, we include an additional cross-attention layer after each self-attention block in the ViT and each cross-attention block in the triplane decoder.

## 3.4    TRAINING AND INFERENCE

**Training**. During the training phase, we uniformly sample time steps $t$ within the range $[1, T]$, and add noise according to a cosine schedule. We sample input images with random camera poses. We also randomly sample additional novel viewpoints to supervise the renderings (as discussed in Sec. 3.1) for better quality. We minimize the following training objective with conditional signal $y$:

$$\mathrm{L} = \mathbb{E}_{t \sim U[1,T], (\mathbf{I}, \boldsymbol{c}) \sim (\mathcal{I}_{full}, \mathcal{C}_{full})} \; \ell\big(\mathbf{I}, \mathrm{R}(\mathrm{E}(\mathcal{I}_t, t, \mathcal{D}, y), \boldsymbol{c})\big) \tag{4}$$

For the image reconstruction loss $\ell(\cdot, \cdot)$, we use a combination of L2 loss and LPIPS loss (Zhang et al., 2018), with loss weights being 1 and 2, respectively.

**Inference**. For inference, we select four viewpoints that uniformly surround the object in a circle to ensure a good coverage of the generated 3D assets. We fix the camera Field-of-Views to 50 degrees for the four views. Since we predict triplane NeRF aligned with the conditioning image's camera frame, we also fix the conditioning image's camera extrinsics to have identity orientation and $(0, -2, 0)$ position, following the practice of LRM (Hong et al., 2023). We output the triplane NeRF from the final denoising step as the generated 3D model. We utilize DDIM (Song et al., 2020a) algorithm to improve the inference speed.

## 4    EXPERIMENTS

In this section, we present an extensive evaluation of our method. In particular, we briefly describe our experiment settings (Sec. 4.1), compare our results with previous works (Sec. 4.2), and show additional analysis and ablation studies (Sec. 4.3).

## 4.1    SETTINGS

**Implementation details**. We use AdamW optimizer to train our model with an initial learning rate of $4e^{-4}$. We also apply a warm-up of $3K$ steps and a cosine decay on the learning rate. We train our denoiser with $256 \times 256$ input images and render $128 \times 128$ image crops for supervision. To save GPU memory for NeRF rendering, we use the deferred back-propagation technique (Zhang et al., 2022). Our final model is a large transformer with 44 attention layers (counting all the self- and cross-attention layers in the encoder and decoder) outputting $64 \times 64 \times 3$ triplanes with 32 channels. We use 128 NVIDIA A100 GPUs to train this model with a batch size of 8 per GPU for $100K$ steps, taking about 7 days. Since the final model takes a lot of resources, it is impractical for us to evaluate the design choices with this large model for our ablation study. Therefore, we also train a small model that consists of 36 attention layers to conduct our ablation study. The small model is trained with 32 NVIDIA A100 GPUs for $200K$ steps (4 days). Please refer to Tab. 6 in the appendix for an overview of the hyper-parameter settings.

**Datasets**. Our model requires only multi-view posed images to train. We use rendered multi-view images of $\sim$730k objects from the Objaverse (Deitke et al., 2023) dataset. For each object, we render 32 images under uniform lighting at random viewpoints with a fixed $50°$ FOV, following the settings of LRM (Hong et al., 2023). To train our text-to-3D model, we use the object captions provided by Cap3D (Luo et al., 2023), which covers a subset of $\sim$660k objects. For image-conditioned (single-view reconstruction) model, we combine the Objaverse data with additional real captures of $\sim$220k objects from the MVImgNet (Yu et al., 2023) dataset, enhancing our model's generalization to out-of-domain inputs (see Fig. 7). We preprocess the MVImgNet dataset in the same way as

Table 1: Evaluation Metrics of single-image 3D reconstruction on ABO and GSO datasets.

| | ABO dataset | | | | | GSO dataset | | | | |
|---|---|---|---|---|---|---|---|---|---|---|
| | FID ↓ | CLIP ↑ | PSNR ↑ | LPIPS ↓ | CD ↓ | FID ↓ | CLIP ↑ | PSNR ↑ | LPIPS ↓ | CD ↓ |
| Point-E | 112.29 | 0.806 | 17.03 | 0.363 | 0.127 | 123.70 | 0.741 | 15.60 | 0.308 | 0.099 |
| Shap-E | 79.80 | 0.864 | 15.29 | 0.331 | 0.097 | 97.05 | 0.805 | 14.36 | 0.289 | 0.085 |
| Zero-1-to-3 | 31.59 | 0.927 | 17.33 | 0.194 | — | 32.44 | 0.896 | 17.36 | 0.182 | — |
| One-2-3-45 | 190.81 | 0.748 | 12.00 | 0.514 | 0.163 | 139.24 | 0.713 | 12.42 | 0.448 | 0.123 |
| Magic123 | 34.93 | 0.928 | 18.47 | 0.180 | 0.136 | 34.06 | 0.901 | 18.68 | 0.159 | 0.113 |
| Ours (S) | 36.77 | 0.915 | 22.62 | 0.194 | 0.059 | 35.16 | 0.888 | 21.80 | 0.150 | 0.046 |
| Ours | **27.88** | **0.949** | **24.15** | **0.127** | **0.046** | **30.01** | **0.928** | **22.57** | **0.126** | **0.040** |

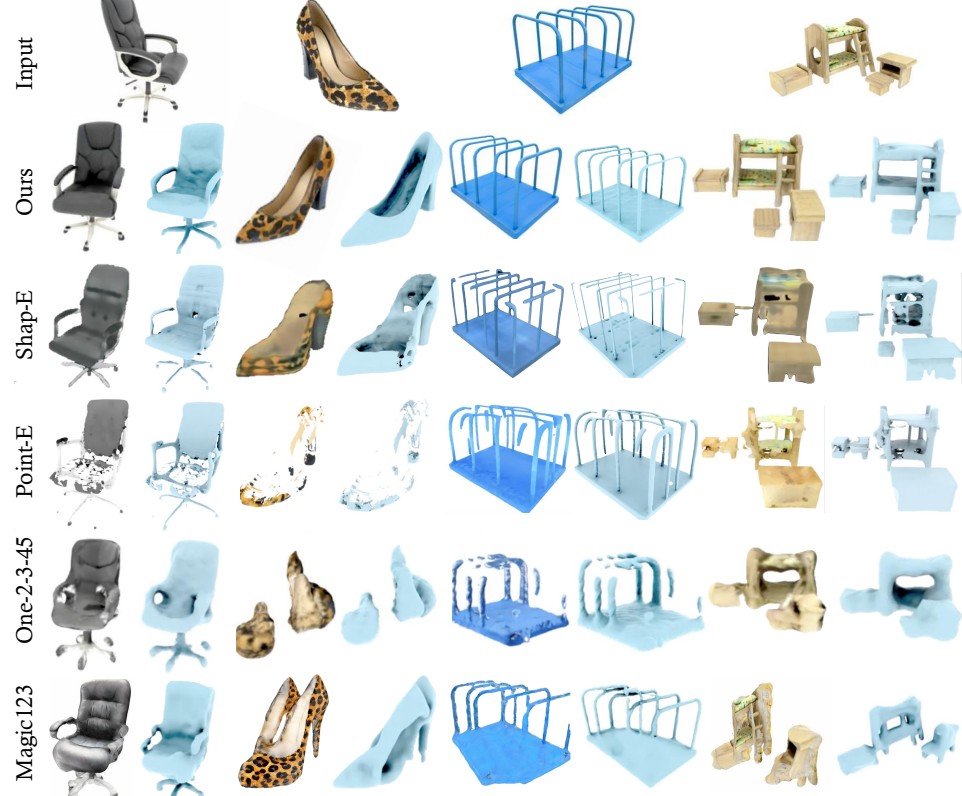

Figure 4: **Qualitative comparisons on single-image reconstruction.**

LRM (Hong et al., 2023): for each capture, we crop out the object of interest for all views, remove the background, and normalize the cameras to tightly fit the captured object into the box $[-1, 1]^3$. In general, these datasets contain a large variety of synthetic and real objects, allowing us to train a generic category-free 3D generative model.

We evaluate our image-conditioned model on novel synthetic datasets, including 100 objects from the Google Scanned Object (GSO) (Downs et al., 2022) and 100 objects from the Amazon Berkeley Object (ABO) (Collins et al., 2022) datasets. This allows for direct comparison of single-view reconstruction with the groundtruth. For each object, we select 20 views that uniformly cover an object from the upper hemisphere to compute metrics; we pick a slightly skewed side view as input.

## 4.2 RESULTS AND COMPARISONS

**Single-image reconstruction.** We compare our image-conditioned model with previous methods, including Point-E (Nichol et al., 2022), Shap-E (Jun & Nichol, 2023), Zero-1-to-3 (Liu et al., 2023b), One-2-3-45 (Liu et al., 2023a), and Magic123 (Qian et al., 2023), on single-image reconstruction. We evaluate the novel-view rendering quality from all methods using PSNR, LPIPS (Zhang et al., 2018), CLIP similarity score (Radford et al., 2021) and FID (Heusel et al., 2017), computed between

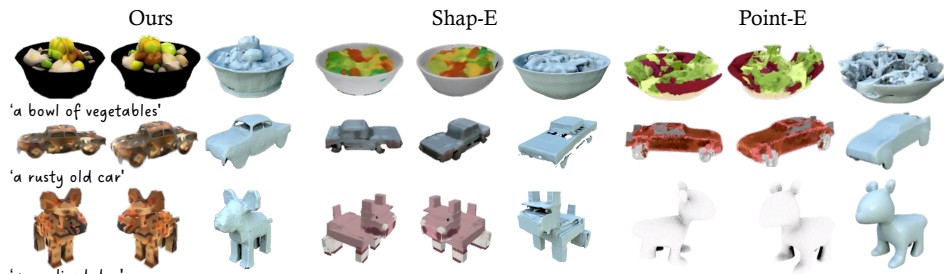

Figure 5: **Qualitative comparisons on Text-to-3D.**

the rendered and GT images. In addition, we also compute the Chamfer distance (CD) for geometry evaluation, for which we use marching cubes to extract meshes from NeRFs. Note that accurate quantitative evaluation of 3D generation remains a challenge in the field due to the generative nature of this problem; we use the most applicable metrics from earlier works to assess our model and baselines.

Tab. 1 reports the quantitative results on the GSO and ABO testing sets respectively. Note that our models (even ours (S)) can outperforms all baseline methods, achieving the best scores across all metrics for both datasets. Our high generation quality is reflected by the qualitative results shown in Fig. 4; our model generates realistic results with higher-quality geometry and sharper appearance details than all baselines.

In particular, the two-stage 3D DMs, Shap-E (3D encoder + latent diffusion) and Point-E (point diffusion + points-to-SDF regression), lead to lower-quality 3D assets, often with incomplete shapes and blurry textures; this suggests the inherent difficulties in denoising 3D points or pretrained 3D latent spaces, a problem our model avoids. On the other hand, Zero-1-to-3 leads to better quantitative results than Shap-E and Point-E on appearnce, because it's a 2D diffusion model finetuned from the pretrained Stable Diffusion (Rombach et al., 2022b) to generate novel-view images. However, Zero-1-to-3 alone cannot output a 3D model needed by many 3D applications and their rendered images suffer from severe inconsistency across viewpoints. This inconsistency also leads to the low reconstruction and rendering quality from One-2-3-45, which attempts to reconstruct meshes from Zero-1-to-3's image outputs. On the other hand, the per-asset optimization-based method Magic123 can achieve rendering quality comparable to Zero-1-to-3 while offering a 3D mdoel. However, these methods require long (hours of) optimization time and also often suffer from unrealistic Janus artifacts (see the high heels object in Fig. 4). In contrast, our approach is a single-stage model with 2D image training objectives and directly generates a 3D NeRF model (without per-asset optimization) while denoising multi-view diffusion. Our scalable model learns strong data priors from massive training data and produces realistic 3D assets without Janus artifacts. In general, our approach leads to fast 3D generation and state-of-the-art single-image 3D reconstruction results.

**Text-to-3D.** We also evaluate our text-to-3D generation results and compare with 3D diffusion models Shap-E (Jun & Nichol, 2023) and Point-E (Nichol et al., 2022), that are also category-agnostic and support fast direct inference. For this experiment, we use Shap-E's 50 text prompts for the generation, and evaluate the results with CLIP precisions (Jain et al., 2022) and averaged precision using two different ViT models, shown in Tab. 2. From the table, we can see that our model achieves the best precision. We also

Table 2: Evaluation Metrics on Text-to-3D.

| Method | VIT-B/32 | | ViT-L/14 | |
|---|---|---|---|---|
| | R-Prec | AP | R-Prec | AP |
| Point-E | 33.33 | 40.06 | 46.4 | 54.13 |
| Shap-E | 38.39 | 46.02 | 51.40 | 58.03 |
| Ours | **39.72** | **47.96** | **55.14** | **61.32** |

show qualitative results in Fig. 5, in which our results clearly contain more geometry and appearance details and look more realistic than the compared ones.

## 4.3 ANALYSIS, ABLATION, AND APPLICATION

We analyze our image-conditioned model and verify our design choices using our small model architecture for better energy efficiency. Refer to Tab. 6 in the appendix for an overview of the hyper-parameter settings for this small model.

Table 3: Ablation on GSO dataset (DMV3D-S).

| #Views | FID ↓ | CLIP ↑ | PSNR ↑ | SSIM ↑ | LPIPS ↓ | CD ↓ |
|---|---|---|---|---|---|---|
| 4 (Ours) | **35.16** | 0.888 | **21.798** | 0.852 | 0.150 | 0.0459 |
| 1 | 70.59 | 0.788 | 17.560 | 0.832 | 0.304 | 0.0775 |
| 2 | 47.69 | 0.896 | 20.965 | 0.851 | 0.167 | 0.0544 |
| 6 | 39.11 | **0.899** | 21.545 | **0.861** | **0.148** | **0.0454** |
| *w.o* Novel | 102.00 | 0.801 | 17.772 | 0.838 | 0.289 | 0.185 |
| *w.o* Plucker | 43.31 | 0.883 | 20.930 | 0.842 | 0.185 | 0.0505 |

| Input | Novel-view | Input | Novel-view |
|---|---|---|---|

Figure 6: **Robustness to out-of-domain inputs**: synthetic (top left), real (bottom left, top right), and generated images (bottom right).

**#Views.** We show quantitative and qualitative comparisons of our models trained with different numbers (1, 2, 4, 6) of input views in Tab. 3 and Fig. 8. We can see that our model consistently achieves better quality when using more images, benefiting from capturing more shape and appearance information. However, the performance improvement of 6 views over four views is marginal, where some metrics (like PSNR, FID) from the 4-view model is even better. We therefore use four views as the default setting to generate all of our main results.

**Multiple instance generation.** Similar to other DMs, our model can generate various instances from the same input image with different random seeds as shown in Fig. 1, demonstrating the diversity of our generation results. In general, we find the multiple instance results can all reproduce the frontal input view while containing varying shape and appearance in the unseen back side.

**Input sources.** Our model is category-agnostic and generally works on various input sources as shown in many previous figures (Fig. 1,2,4). We show additional results in Fig. 6 with various inputs out of our training domains, including synthetic renderings, real captures, and generated images. Our method can robustly reconstruct the geometry and appearance of all cases.

**Ablation of MVImgNet.** We compare our models trained with and without the real MVImgNet dataset on two challenging examples. As shown in Fig. 7, we can see that the model without MVImgNet can lead to unrealistic flat shapes, showcasing the importance of diverse training data.

**Ablations of novel-view supervision and Plucker rays.** We compare with our ablated models including one trained without the novel-view supervision, and one without the Plucker ray conditioning (using the *adaLN-Zero block* conditioning instead). We can also see that the novel view rendering supervision is critical for our model. Without it, all quantitative scores drop by a large margin due to that the model cheats by pasting the input images on view-aligned planes instead of reconstructing plausible 3D shapes. In addition, our design of Plucker coordinate-based camera conditioning is also effective, leading to better quantitative results than the ablated model.

**Application.** The flexibility and generality of our method can potentially enable broad 3D applications. One useful image editing application is to lift any objects in a 2D photo to 3D by segment them (using methods like SAM (Kirillov et al., 2023)) and reconstruct the 3D model with our method, as shown in Fig. 1 and 2.

## 5 CONCLUSION

We present a novel single-stage diffusion model for 3D generation which generates 3D assets by denoising multi-view image diffusion. Our multi-view denoiser is based on a large transformer model (Hong et al., 2023), which takes noisy multi-view images to reconstruct a clean triplane NeRF, outputting denoised images through volume rendering. Our framework supports text- and image-conditioning inputs, achieving fast 3D generation via direct diffusion inference without per-asset optimization. Our method outperforms previous 3D diffusion models for text-to-3D generation and achieves state-of-the-art quality on single-view reconstruction on various testing datasets.

**Ethics Statement.**  Our generative model is trained on the Objaverse data and MVImgNet data. The dataset (about 1M) is smaller than the dataset in training 2D diffusion models (about 100M to 1000M). The lack of data can raise two considerations. First, it can possibly bias towards the training data distribution. Secondly, it might not be powerful enough to cover all the vast diversity in testing images and testing texts. Our model has certain generalization ability but might not cover as much modes as the 2D diffusion model can. Given that our model does not have the ability to identify the content that is out of its knowledge, it might introduce unsatisfying user experience. Also, our model can possibly leak the training data if the text prompt or image input highly align with some data samples. This potential leakage raises legal and security considerations, and is shared among all generative models (such as LLM and 2D diffusion models).

**Reproducibility Statement.**  We provide detailed implementation of our training method in the main text and also provide the model configurations in Table 6 of the appendix. We will help resolve uncertainty of our implementation in open discussions.

**Acknowledgement.**  We would like to thank Nathan Carr, Duygu Ceylan, Paul Guerrero, Chun-Hao Huang, and Niloy Mitra for discussions about this project. We also thank Yuan Liu for providing testing images from Syncdreamer.

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

## A APPENDIX

### A.1 ROBUSTNESS EVALUATION.

We evaluate our model on GSO (Downs et al., 2022) renderings that use different camera Field-Of-Views (FOVs) and lighting conditions to justify its robustness. Specifically, while the MVImgNet dataset include diverse camera FOVs and lighting conditions, the Objaverse renderings we are also trained on share a constant 50° FOV and uniform lighting. We evaluate the robustness of our image-conditioned model by testing images with other FOV angles and complex environmental lightings. As shown in Tab. 4, our model is relatively robust to the FOV of the captured images, though quality indeed drops when the actual FOV deviates more from the 50° FOV we assume during inference (see Sec. 3.4). However, it exhibits lower sensitivity to lighting variations, leading to similar quality across different lighting conditions. When the lighting is non-uniform, our model bakes the shading effects into the NeRF appearance, yielding plausible renderings.

Table 4: Robustness on GSO dataset.

| Lighting/Fov | Appearance | | | | | Geometry |
|---|---|---|---|---|---|---|
| | FID ↓ | CLIP ↑ | PSNR ↑ | SSIM ↑ | LPIPS ↓ | CD ↓ |
| Ours | **30.01** | **0.928** | **22.57** | **0.845** | **0.126** | **0.0395** |
| Fov10 | 35.69 | 0.912 | 19.136 | 0.820 | 0.207 | 0.0665 |
| Fov30 | 32.309 | 0.921 | 20.428 | 0.839 | 0.166 | 0.0527 |
| Fov70 | 32.095 | 0.921 | 20.961 | 0.860 | 0.154 | 0.0616 |
| Fov90 | 34.438 | 0.912 | 19.952 | 0.855 | 0.190 | 0.0754 |
| city | 33.31 | 0.916 | 21.19 | 0.831 | 0.142 | 0.0437 |
| night | 36.32 | 0.907 | 20.383 | 0.829 | 0.161 | 0.0413 |
| sunrise | 33.264 | 0.917 | 21.080 | 0.843 | 0.140 | 0.0423 |
| studio | 36.32 | 0.927 | 21.383 | 0.839 | 0.141 | 0.0428 |

### A.2 QUANTATIVE EVALUATION ON MVIMGNET.

MVImgNet (Yu et al., 2023) contains a diverse set of real data, which helps improve our generalization capabilities for real data or out-of-domain data, as demonstrated in Fig 7. We also perform quantative evaluation on the model with and without MVImgNet on the GSO dataset (Downs et al., 2022) in Tab. 5. The reconstructed results in terms of appearance and geometry are similar to the previous results only trained with Objaverse, indicating that MVImgNet improves generalization without compromising the quality of reconstruction. We train both settings for an equal number of 100K iterations with exactly the same learning rate schedules and computes.

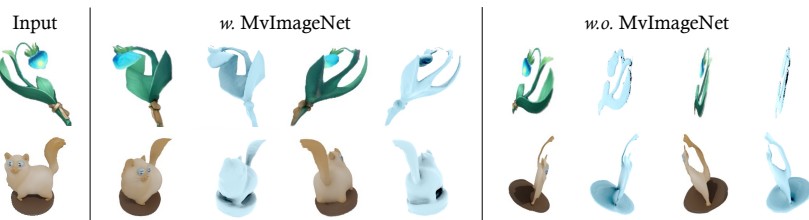

| Input | w. MvImageNet | | | w.o. MvImageNet | | |

Figure 7: **Qualitative comparison of our model trained with and without MVImgNet.**

Table 5: Ablation of MVImgNet.

| #Views | Appearance | | | | | Geometry |
|---|---|---|---|---|---|---|
| | FID ↓ | CLIP ↑ | PSNR ↑ | SSIM ↑ | LPIPS ↓ | CD ↓ |
| w. MvImageNet | **30.01** | **0.928** | **22.57** | 0.845 | **0.126** | 0.0395 |
| w.o MvImageNet | 27.76 | 0.924 | 21.85 | **0.850** | 0.128 | **0.0378** |

Our experiments are implemented in the PyTorch and the codebase is built upon guided diffusion (Dhariwal & Nichol, 2021). For the AdamW optimizer, we use a weight-decay $0.05$ and beta $(0.9, 0.95)$. Table 6 presents the detailed configuration of our various image-conditioned models. The architecture of the text-conditioned model closely mirrors that of the image-conditioned models, with the primary distinction being the approach to injecting the condition signal. For text-conditioned models, we employ the CLIP text encoder to derive text embeddings, integrating them into our denoiser through cross-attention layers. Specifically, in each transformer block within the encoder and decoder, a new cross-attention layer is introduced between the original attention and FFN. In such a case, text-conditioned models consistently exhibit larger sizes than their image-conditioned counterparts, resulting in a slightly slower inference speed. During inference, we adopt a classifier-free guidance approach Ho & Salimans (2022) with a scale of 5 to generate 3D assets conditioned on text.

|  |  | Small | Large |
|---|---|---|---|
| Encoder | Image resolution | 256×256 | 256×256 |
|  | Patch size | 16 | 8 |
|  | Att. Layers | 12 | 12 |
|  | Att. channels | 768 | 768 |
| Decoder | Triplane tokens | $32 \times 32 \times 3$ | $32 \times 32 \times 3$ |
|  | Att. channels | 768 | 1024 |
|  | Att. layers | 24 (12a+12c) | 32 (16a+16c) |
|  | Triplane upsample | 1 | 2 |
|  | Triplane shape | $32 \times 32 \times 3 \times 32$ | $64 \times 64 \times 3 \times 32$ |
| Renderer | Rendering patch size | 64 | 128 |
|  | Ray-marching steps | 48 | 128 |
|  | MLP layers | 10 | 10 |
|  | MLP width | 64 | 64 |
|  | Activation | ReLU | ReLU |
| Diffusion | Times steps | 1000 | 1000 |
|  | Prediction target | $x_0$ | $x_0$ |
|  | Schedule | cosine | cosine |
| Traininig | Learning rate | 4e-4 | 4e-4 |
|  | Optimizer | AdamW | AdamW |
|  | Warm-up steps | 3000 | 3000 |
|  | Batch size per GPU | 8 | 8 |
|  | #GPUS | 32 | 128 |
|  | Iterations | $200K$ | $100K$ |
|  | Training time | 4 days | 7 days |
| Dataset | Source | MVImgNet & Objaverse | MVImgNet & Objaverse |
|  | Mixing ratio | 1:3 | 1:3 |
|  | Resolution | 256 | 256 |

Table 6: Implementation details for our models. Att. denotes the attention. $a$ and $c$ represents the self-attention and cross attention.

## A.4 VIEW NUMBERS

We have compared the effects of using different numbers of views quantitatively in Tab. 3. Here, we also present qualitative results in Fig. 8. When there is only one view, the predicted novel view is very blurry. However, when the view number increases to four, the results become much clearer. When using six views, the improvement compared to four views is not significant, consistent to the metrics reported in Tab. 3, indicating performance saturation. Therefore, our model uses four views as the default configuration.

## A.5 MORE COMPARISON.

We also include more qualitative comparison on single-view image reconstruction in Fig. 9.

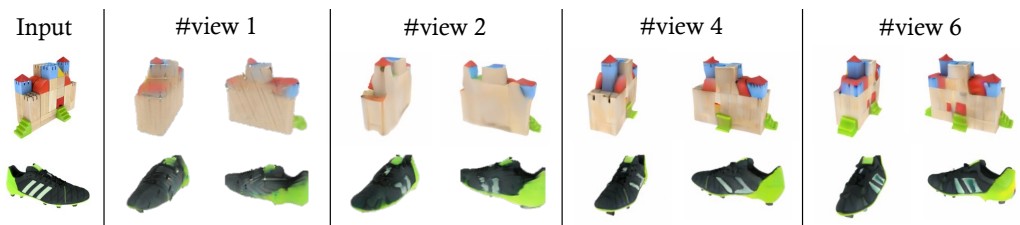

Figure 8: **Qualitative comparison on different view numbers.**

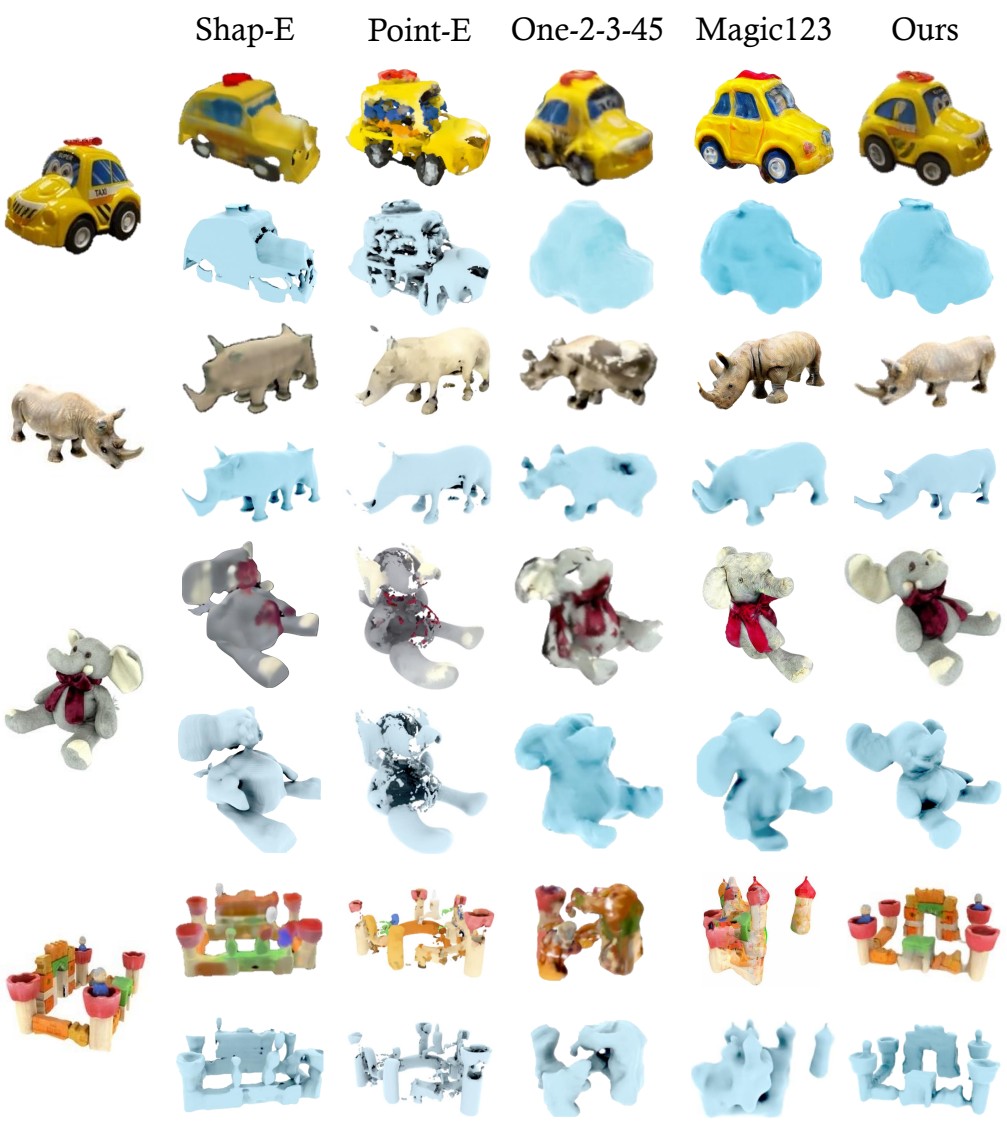

Figure 9: **Qualitative comparison on single-image reconstruction.**

