# OpenReview forum: "DMV3D: Denoising Multi-view Diffusion Using 3D Large Reconstruction Model"
_ICLR.cc/2024/Conference — ICLR 2024 spotlight_

### Official Review · Reviewer_X5Dv · 2023-10-23

**Soundness:** 4 excellent
**Presentation:** 3 good
**Contribution:** 4 excellent
**Rating:** 10
**Confidence:** 5

**Summary:**

The paper proposes DMV3D, a 3D generation approach that uses a transformer-based 3D large reconstruction model to denoise multi-view diffusion. The reconstruction model incorporates a triplane NeRF representation and, functioning as a denoiser, can denoise noisy multi-view images via 3D NeRF reconstruction and rendering, achieving single-stage 3D generation in the 2D diffusion denoising process. The model is trained on large-scale multi-view image datasets of extremely diverse objects using only image reconstruction losses, without accessing 3D assets.

**Strengths:**

1. The first contribution of this paper is to scale 3D diffusion generative models to very diverse categories and objects. Previous models such as DiffRF, etc. can only generalize within some shapenet like datasets with no more than 13 categories.

2. The model demonstrates a novel method to conduct multiview diffusion. Instead of build attentions across views like mvdreamer or syncdreamer, they use attention to attend with learnable triplane tokens and with each other, therefore incorporating the 3D spatial prior in the process.

3. The model shows good results of 3d generation, especially high quality geometry, which alwyas fail in SDS or nerf2nerf lines of works.

**Weaknesses:**

1.It seems the model learns from the objaverse and mvimagnet, which contain mostly single objects or separated objects. even the examples in out of domain results, in figure 6, the objects are not complicated as people use in SD-based models.

**Questions:**

1. As mentioned in weakness 1,  I would like to see some results of "bunny seating on pancake", this kind of generation. Even it is hard to do text to 3d, since the training set doesn't have compound objects, is it possible to do 2d conditioned 3d generation with this kind of prompt?

2. The author mentioned in the 2d conditioned 3d generation task, they do not add noise to the reference view, however, some of other diffusion models usually also add noise to the reference view and each step, use the gt x0 of the that view and add new noise in ancestral sampling. The logic behind is the model is trained with noise images paired with the corresponding time step embedding, the clean image strategy will shock the model in inference. I wonder, in inference, if this clean ref image strategy can bring benefit over adding noise from x0.

---

> ### Author Response · Authors · 2023-11-20
> **Official Comment by Authors**
>
> We thank the reviewer for your appreciation of our method’s capability to handle diverse objects, our novelty to use a reconstructor as a multi-view denoiser, and our high-quality results. We address your questions below.
>
> 1. **Complex prompts**: our text-to-3D model has limited capability to handle complex prompts due to the limited scale of multi-view data. But as the reviewer points out, we can use Stable Diffusion to generate a single image from a complex prompt, and then use our image-conditioned model to lift it into 3D. We show the results here (https://dmv3d.github.io/rebuttal_sdcond.html).
>
> 2. **Noise in reference image**: we tried adding noise to reference images, but found that this is essentially equivalent to unconditional generation, causing the model to less respect the respect image. Hence, we choose to use the noise-free reference image across all timesteps.

---

> > ### Comment · Reviewer_X5Dv · 2023-11-20
> >
> > I'm in general satisfied with the authors' answers which address all my concerns. I would like to raise my score.

---

> ### Author Response · Authors · 2023-11-21
> **Official Comment by Authors**
>
> Dear Reviewer,
>
> We sincerely thank the reviewer for your feedback and your appreciation of our methods and results.
> And we would like to express our gratitude for raising the score of our paper!

---

### Official Review · Reviewer_PYdg · 2023-10-30

**Soundness:** 3 good
**Presentation:** 3 good
**Contribution:** 3 good
**Rating:** 8
**Confidence:** 4

**Summary:**

The paper presents a method to generate novel 3D objects based on a diffusion model that encloses a large reconstruction model. To leverage the strong generative power of 2D models while improving the 3D consistency of the generated objects, the diffusion model operates on the domain of multi-view images and internally learns a transformer-based reconstruction model to build a 3D representation, which is later rendered into denoised output images. In order for the model to generalize across different categories, the reconstruction model uses the DINO features to bootstrap the features used for deriving tokens. Results show that by training jointly on the Objaverse dataset and MVImageNet dataset, the model is able to generate diverse shapes, conditioned either on images or texts. The usage of the transformer-based large reconstruction model improves the 3D consistency while maintaining a good generation quality.

**Strengths:**

- The paper is clearly written and well presented. The notations are clear and the illustrations are informative. It's easy to read and understand most of the technical details and design choices.
- Though conceptually similar to, e.g., MVDiffusion and RenderDiffusion (or diffusion with forward models), the method elegantly combines the advantages of both works with the help of a generalizable large reconstruction model using transformers, hence lifting the previous constraints within only one single category.
- The method naturally enables conditioning over images by fixing the diffusion variables, leading to a new scheme for bridging 2D and 3D domains.
- The generated shapes are of high quality and surpass the baselines by a considerable margin.

**Weaknesses:**

- In contrast to Image-based diffusion models, the runtime efficiency might be compromised since the reconstruction model operates during every iteration of sampling. Investigating whether the reconstruction can be repurposed or distributed over the intermediate denoising phases could be insightful, especially since the current intermediate reconstructed model is discarded (would be great if they could be visualized), leading to potential wastage.

- The textures produced lack sharpness. Exploring the proposed framework's performance on higher-resolution images and 3D triplanes would be intriguing. Additionally, employing a hybrid representation that decouples geometry and textures could yield enhanced results.

**Questions:**

- How the camera viewpoints are sampled during the training process? Would the reconstruction model easily fall into a local minima where the 3D results become trivial by generating planes that are parallel to the image planes?

**Details Of Ethics Concerns:**

Not applicable.

---

> ### Author Response · Authors · 2023-11-20
> **Official Comment by Authors**
>
> We thank the reviewer for your appreciation of our clear presentation, method’s elegance and high-quality results. We address your concerns and questions below.
>
> 1. **Runtime efficiency**: we agree with the reviewer that it might be a good idea to interleave geometry-based multi-view denoising and geometry-free multi-view denoising for best runtime efficiency, as rendering 4 views from a triplane at each denoising step does have a negative influence on the inference speed. We leave this point for future exploration. We also visualize the intermediate reconstructed results during denoising process at this link (https://dmv3d.github.io/rebuttal_x0vis.html).
>
> 2. **Blurry texture**: we agree with the reviewer that our generated textures for unseen parts are a bit blurry at its current stage, although we outperform baselines. For future work, it’s interesting to explore increasing image and triplane resolution and decoupling geometry and texture modeling to address this issue.
>
> 3. **Viewpoint sampling**: we randomly sample 4 views during training to use as our input views, and another randomly selected 4 novel views as supervision. During inference, we sample 4 structured viewpoints around an object to denoise.
>
> 4. **Degenerate solution**: we use novel view supervision to prevent the degenerate solution of generating 4 flat planes to explain the input views (see table 3). The results without novel view supervision can be found at this link (https://dmv3d.github.io/rebuttal_wonovelview.html).

---

> ### Author Response · Authors · 2023-11-21
> **Official Comment by Authors**
>
> Dear reviewer,
>
> We kindly request your feedback on whether our response has addressed your concerns. If you have any remaining questions or concerns, we are more than happy to address them. Thank you for your time and consideration!

---

### Official Review · Reviewer_nAvZ · 2023-11-01

**Soundness:** 3 good
**Presentation:** 3 good
**Contribution:** 2 fair
**Rating:** 6
**Confidence:** 4

**Summary:**

This paper presents an approach to 3D generation via a single-stage diffusion model. By denoising multi-view image diffusion, the authors aim to generate realistic 3D assets. Central to this methodology is a large transformer model that processes multi-view noisy images to reconstruct a clean triplane NeRF, subsequently yielding denoised images through neural rendering. The proposed method showcases flexibility, supporting both text- and image-conditioning inputs, and claims rapid 3D generation without requiring per-asset optimization. The approach is evaluated and shown to be superior to previous 3D diffusion models in certain domains.

**Strengths:**

(+) The paper showcases impressive results in 3D generation compared to prior methods.

(+) The method's ability to accommodate text- and image-conditioning inputs augments its versatility, making it potentially suitable for diverse applications.

(+) The paper is well-structured and clearly explains both the methodology and foundational design choices.

**Weaknesses:**

Although the paper showcases promising results and a solid methodology; however, its level of novelty is unclear:

- It appears that the proposition combines techniques that have been used before. The 3D diffusion part of the proposal seems to have been influenced by "Viewset Diffusion (ICCV 2023)", while the design and training approach of the large-scale transformer model is similar to "LRM: LARGE RECONSTRUCTION MODEL FOR SINGLE IMAGE TO 3D", which was also submitted to ICLR 2024.
- Concerns have been raised about potential overlap with the LRM manuscript, questioning submission singularity.

Besides, a balanced perspective is lacking due to the absence of discussion on the paper's limitations, which could provide valuable insights for potential areas of improvement.

**Questions:**

- Given the inherent training characteristics of diffusion models, how does DMV3D achieve a training timeframe analogous to LRM? It would be helpful if the authors could provide an explanation.
- It is important to clarify the extent of overlap between this work and the LRM submission in order to understand the distinctiveness of the contributions in this manuscript.

---

> ### Author Response · Authors · 2023-11-20
> **Official Comment by Authors**
>
> We thank the reviewer for your appreciation of our impressive results, our method’s great versatility and clear presentation. Please see our responses to your concerns below.
>
> 1. **DMV3D vs LRM**: We would like to clarify the different contributions of LRM and DMV3D. Please note that, in DMV3D, we acknowledge LRM as prior work and have explicitly cited LRM in section 3.2. Our work makes orthogonal contributions,  focusing on different challenges that are not covered by LRM. In particular, LRM focuses on the task of single-image reconstruction only and introduces the first scalable transformer model trained on massive multi-view data. However, in essence, LRM is a deterministic model that cannot model the inherent uncertainty (multiple plausible solutions exist for unseen parts of the 3D objects) in single-image 3D reconstruction. In contrast, DMV3D aims to build a general probabilistic diffusion model for 3D generation. Our framework is not only applicable to single-image reconstruction but also text-to-3D generation – a task completely uncovered by LRM.
>
>     To do so, we propose a novel single-stage multi-view diffusion framework and adopt a transformer model to achieve multi-view denoising. Again, please note that we have cited LRM (by anonymously putting it in the supplemental material) and clearly stated in Sec.3.2 that our transformer architecture is inspired by LRM to avoid any misconception about dual submission. We emphasize that the whole story of our paper, including our entire method section (Sec. 3), does not focus on the original transformer contribution made by LRM. Instead, our focus is mainly on: 1) how to build a multi-view diffusion framework for 3D generation (Sec. 3.1), 2) how to improve/extend the transformer model to support multi-view denoising (Sec. 3.2), and 3) how to enable both text and image conditioning (Sec. 3.3). These aspects are distinct from the scope of LRM. Furthermore, our transformer architecture is also different from the one in LRM in taking noisy multi-view images as input, being conditioned by the diffusion time step, utilizing a new ray-conditioning technique to better handle the diverse camera intrinsics, and supporting text conditioning.
>
>     As a result, even for the same single-image reconstruction problem, DMV3D and LRM address it in very different ways. DMV3D leverages multi-view diffusion denoising, whereas LRM  performs single-view image-to-NeRF translation. The transformer inputs (multi-view noisy images vs a single-view clean image) and the inference processes (DDIM-based multi-step diffusion denoising vs single-step transformer inference) are highly different. Moreover, because of being a probabilistic generative model, DMV3D can generate multiple high-quality 3D assets given the same single image input (as shown in Fig. 1 and our project page), while LRM can only generate one single result. The probabilistic modeling introduced by DMV3D also enhances the realism and reduces the blurriness of generated textures for unseen parts; in contrast, LRM tends to generate blurred results due to mode averaging.
>
>     In sum, different from the deterministic model in LRM, our DMV3D introduces a novel probabilistic model for 3D generation and contributes 1) a novel multi-view diffusion framework 2) the general support for both text and image conditioning and  3) new architecture designs to support multi-view denoising and text/image conditioning. The contributions are made orthogonal to those of LRM and we believe these are valuable contributions to the field towards a general, practical, and high-quality 3D generative model.
>
> 2. **Novelty**: As discussed in the related work, we are inspired by the RenderDiffusion work for using a reconstructor as a denosier. Concurrent work, Viewset Diffusion, adopts the same idea from RenderDiffusion, but was only demonstrated to work on limited data lacking diversity. Our novel transformer-based 3D denoiser architecture is different from the reconstructor architecture in viewset diffusion and enables state-of-the-art results for scalable, diverse, and high-quality 3D generation.
>
> 3. **Limitations**: one limitation with our text-to-3D model is that we cannot handle complex text prompts as well as 2D diffusion models like stable diffusion, as a result of the limited scale of multi-view data compared with 2D data. Another limitation is that the generated textures at unseen views are still a bit blurry compared with input view. It will be interesting to further boost the quality of texture sharpness in future work.
>
> 4. **Training time**: despite our method being a diffusion model, we only sample a single time-step at a time to train our multi-view denoiser. This single-step setting is similar to what LRM does except our input images are noisy; hence our model trains as fast as LRM during training.

---

> > ### Comment · Reviewer_nAvZ · 2023-11-20
> >
> > I would like to express my appreciation to the authors for their detailed and thoughtful rebuttal. In the response, the authors have clarified several important points that address the concerns raised in my original review.
> >
> > First and foremost, I want to reaffirm my understanding and appreciation of the contributions made by the paper, particularly in introducing the multi-view denoising framework for 3D generation. The promising results that were presented in the paper are worth noting. I also note that the authors had made appropriate citations to related anonymous submissions, namely LRM and Instant3D.
> >
> > One of my primary concerns in the original review was regarding the positioning of LRM in relation to the present paper. The authors now clearly acknowledged LRM as prior work in the rebuttal, which alleviates many of my concerns. However, __treating LRM as a prior work also introduces certain expectations__. For instance, when claiming that the proposed method in DMV3D can potentially achieve better results in unseen area reconstruction compared to LRM, it might be beneficial to support such claims with experimental or visual comparison. While the generative formulation in DMV3D seems theoretically capable of addressing the averaging issue in unseen areas, empirical validation would enhance the paper's credibility.
> >
> > In conclusion, I acknowledge that DMV3D offers valuable insights to the research community and addresses many of the concerns raised in the original review, including the overlap with LRM. Given these clarifications and improvements, I am inclined to raise my rating for this submission. I would also like to recommend that the authors consider incorporating the discussion related to LRM into the main paper. Additionally, as the scenario of two closely related submissions within the same conference is relatively uncommon, I would also love to hear the comments from other reviewers.

---

> > > ### Author Response · Authors · 2023-11-21
> > > **Official Comment by Authors**
> > >
> > > We thank the reviewer for your appreciation of our contribution and impressive results. Please see our responses to your concerns about the comparison with LRM below.
> > >
> > > 1. **Qualitative and Qualitative Comparison with LRM**:  We conducted a qualitative and quantitative comparison with LRM with the same evaluation settings in Sec 4.2 on GSO dataset. As shown in the Table, our model outperforms LRM in all evaluation metrics related to appearance and geometry.  This superiority stems from the fact that LRM is a deterministic model that struggles to capture the distribution of unseen parts, resulting in blurry textures.
> > >
> > >     Additionally, we also include visual comparisons across various samples, further supporting that our probabilistic generative model produces better visual samples from unseen viewpoints compared to deterministic methods like LRM.  (The link is at https://dmv3d.github.io/rebuttal_complrm.html)
> > >
> > > |    | FID $\downarrow$ | CLIP-Similarity $\uparrow$ | PSNR $\uparrow$ |  LPIPS $\downarrow$ |  Chamfer Distance $\downarrow$ |
> > > | :---                |   :----:       |    :----:   |      :----:   |     :----:   |    ---: |
> > > | LRM   | 31.44  | 0.902  |  19.60  |  0.163  |  0.053  |
> > > | Ours  | **30.01** | **0.928** | **22.57** | **0.126** | **0.040** |
> > >
> > > 2. **Update Paper**: Thank you for your valuable suggestions. We have incorporated the discussion with LRM into our introduction and related work section, and the updated texts have been highlighted in red. Additionally, we have included a comparison with LRM in the Appendix. The paper is updated and you can use the 'view-difference' functionality provided by OpenReview for comparisons.

---

> > > > ### Author Response · Authors · 2023-11-22
> > > > **Official**
> > > >
> > > > Dear reviewer, please kindly let us know if our response has addressed your concerns with the approach of the rebuttal deadline. We are happy to answer any of your remaining concerns if you have any. We would also appreciate your response regarding whether you might be willing to raise your score. Thank you!

---

### Official Review · Reviewer_3SCV · 2023-11-04

**Soundness:** 3 good
**Presentation:** 3 good
**Contribution:** 3 good
**Rating:** 8
**Confidence:** 5

**Summary:**

The paper proposes a 3D generation method that uses a transformer-based 3D large reconstruction model to denoise multi-view diffusion. The proposed method supports both text- and image-conditioned 3D generation. Experimental results seem promising.

**Strengths:**

The idea of directly denoising a triplane-based NeRF is interesting. The result of multi-view diffusion is promising.

**Weaknesses:**

1. Does the method use a pre-trained stable diffusion model or train the DDPM from scratch? If from scratch, how is the generalization ability guaranteed?

2. For multi-view diffusion, what is the number of views for training and inference?

**Questions:**

Please see the weakness above.

---

> ### Author Response · Authors · 2023-11-20
> **Official Comment by Authors**
>
> We thank the reviewer for your appreciation of our method and result. We address your questions below.
>
> 1. Train from scratch: all of our models are trained from scratch without using pretrained diffusion models. Generalization of our model is backed up by large-scale training on the massive multi-view data including Objaverse and MvImgNet (See Fig. 7).
>
> 2. Number of views: we use 4 views during training and inference (see section 4.3).

---

> ### Author Response · Authors · 2023-11-21
> **Official Comment by Authors**
>
> Dear reviewer,
>
> We kindly request your feedback on whether our response has addressed your concerns. If you have any remaining questions or concerns, we are more than happy to address them. Thank you for your time and consideration!

---

> > ### Author Response · Authors · 2023-11-22
> > **Official Comment by Authors**
> >
> > Dear reviewer, please kindly let us know if our response has addressed your concerns with the approach of the rebuttal deadline. We are happy to answer any of your remaining concerns if you have any. Thank you!

---

> > > ### Comment · Reviewer_3SCV · 2023-11-22
> > >
> > > Thanks for the response. There are still some concerns compared to some concurrent works.
> > > 1. The technical contribution of this work. Compared to LRM, this paper wraps the triplane-NeRF into a diffusion process. What is the motivation for this choice?
> > > 2. In terms of multi-view consistency, it is also beneficial to compare with SyncDreamer [1], which seems very competitive in consistency.
> > >
> > > [1] SyncDreamer: Generating Multiview-consistent Images from a Single-view Image, arXiv, Sep 7.

---

> > > > ### Author Response · Authors · 2023-11-23
> > > >
> > > > We thank the reviewer for your response; please find our response below:
> > > >
> > > > **DMV3D vs LRM**: Our work makes orthogonal contributions to LRM, focusing on different challenges that are not covered by LRM. In particular, LRM focuses on the task of single-image reconstruction only and introduces the first scalable transformer model trained on massive multi-view data. However, in essence, LRM is a deterministic model that cannot model the inherent uncertainty (multiple plausible solutions exist for unseen parts of the 3D objects) in single-image 3D reconstruction. In contrast, DMV3D aims to build a general probabilistic diffusion model for 3D generation. Our framework is not only applicable to single-image reconstruction but also text-to-3D generation – a task completely uncovered by LRM.
> > > >
> > > > To do so, we propose a novel single-stage multi-view diffusion framework and adopt a transformer model to achieve multi-view denoising. We emphasize that the whole story of our paper, including our entire method section (Sec. 3), does not focus on the original transformer contribution made by LRM. Instead, our focus is mainly on: 1) how to build a multi-view diffusion framework for 3D generation (Sec. 3.1), 2) how to improve/extend the transformer model to support multi-view denoising (Sec. 3.2), and 3) how to enable both text and image conditioning (Sec. 3.3). These aspects are distinct from the scope of LRM. Furthermore, our transformer architecture is also different from the one in LRM in taking noisy multi-view images as input, being conditioned by the diffusion time step, utilizing a new ray-conditioning technique to better handle the diverse camera intrinsics, and supporting text conditioning. As a result, even for the same single-image reconstruction problem, DMV3D and LRM address it in very different ways. DMV3D leverages multi-view diffusion denoising, whereas LRM performs single-view image-to-NeRF translation. The transformer inputs (multi-view noisy images vs a single-view clean image) and the inference processes (DDIM-based multi-step diffusion denoising vs single-step transformer inference) are highly different. Moreover, because of being a probabilistic generative model, DMV3D can generate multiple high-quality 3D assets given the same single image input (as shown in Fig. 1 and our project page), while LRM can only generate one single result. The probabilistic modeling introduced by DMV3D also enhances the realism and reduces the blurriness of generated textures for unseen parts; in contrast, LRM tends to generate blurred results due to mode averaging.
> > > >
> > > > In sum, different from the deterministic model in LRM, our DMV3D introduces a novel probabilistic model for 3D generation and contributes 1) a novel multi-view diffusion framework 2) the general support for both text and image conditioning and 3) new architecture designs to support multi-view denoising and text/image conditioning. The contributions are made orthogonal to those of LRM and we believe these are valuable contributions to the field towards a general, practical, and high-quality 3D generative model.
> > > >
> > > > We also conducted a qualitative and quantitative comparison with LRM with the same evaluation settings in Sec 4.2 on GSO dataset. As shown in the Table, our model outperforms LRM in all evaluation metrics related to appearance and geometry. This superiority stems from the fact that LRM is a deterministic model that struggles to capture the distribution of unseen parts, resulting in blurry textures. Additionally, we also include visual comparisons across various samples, further supporting that our probabilistic generative model produces better visual samples from unseen viewpoints compared to deterministic methods like LRM. (The link is at https://dmv3d.github.io/rebuttal_complrm.html)
> > > >
> > > > |    | FID $\downarrow$ | CLIP-Similarity $\uparrow$ | PSNR $\uparrow$ |  LPIPS $\downarrow$ |  Chamfer Distance $\downarrow$ |
> > > > | :---                |   :----:       |    :----:   |      :----:   |     :----:   |    ---: |
> > > > | LRM   | 31.44  | 0.902  |  19.60  |  0.163  |  0.053  |
> > > > | Ours  | **30.01** | **0.928** | **22.57** | **0.126** | **0.040** |
> > > >
> > > > **Comparison with SyncDreamer**: we agree with the reviewer that it is interesting to compare our work with the concurrent SyncDreamer. But we respectfully disagree that not comparing with SyncDreamer will be a valid concern hurting our submission, as SyncDreamer is submitted to arXiv on Sept. 7, while our manuscript is submitted to OpenReview on Sept. 28. With such a short time difference, we are not obligated to do the comparison according to ICLR reviewer guideline: https://iclr.cc/Conferences/2024/ReviewerGuide . This said, we are happy to include a comparison in our final draft upon acceptance.

---

> > > > > ### Author Response · Authors · 2023-11-23
> > > > >
> > > > > Dear reviewer, we just made a qualitative comparison with SyncDreamer here (https://dmv3d.github.io/rebuttal_comsyncdreamer.html). SyncDreamer can generate view-inconsistent multi-views, while our generated multi-views are view-consistent by formulation as they are renderings from the same reconstructed NeRF. Please let us know if you have other concerns.

---

### Meta-Review · Area_Chair_CtxA · 2023-12-13

**Metareview:**

This paper proposes DMV3D, a 3D generation approach that uses a transformer-based 3D large reconstruction model to denoise multi-view diffusion. It incorporates a triplane NeRF representation and, functioning as a denoiser, can denoise noisy multi-view images via 3D NeRF reconstruction and rendering, achieving single-stage 3D generation in the 2D diffusion denoising process. The authors train DMV3D on large-scale multi-view image datasets of extremely diverse objects using only image reconstruction losses, without accessing 3D assets. They demonstrate state-of-the-art results for the single-image reconstruction and also show high-quality text-to-3D generation results outperforming previous 3D diffusion models.

(a) Strengths of the paper

The paper demonstrates impressive results in 3D generation, showing advancements over previous methods (Reviewer X5Dv). It's commended for its ability to handle both text- and image-conditioning inputs, showing versatility (Reviewer 3SCV). The structure is well-articulated, explaining the methodology and foundational design choices clearly, and the results reflect high-quality geometry (Reviewer PYdg).

(b) Weaknesses of the paper

The novelty level is uncertain, with some techniques appearing to be derivatives of existing methods, raising questions about originality (Reviewer nAvZ). Concerns about the model's efficiency and the sharpness of generated textures indicate room for optimization (Reviewer PYdg). There's also a need for clarity on how the DMV3D relates to other models, and the specific contributions of the proposed method could be articulated more distinctly (Reviewer nAvZ).

Given the significant strengths and the potential impact of the paper, alongside the constructive feedback provided by the reviewers, it is suggested for acceptance as a spotlight.

**Justification For Why Not Higher Score:**

This paper needs further clarity on its novelty and the relationship with existing methods, as highlighted by Reviewer nAvZ. Furthermore, questions about the efficiency of the model and texture quality, as noted by Reviewer PYdg, suggest that there are areas for optimization.

**Justification For Why Not Lower Score:**

The model has shown to outperform existing methods, as acknowledged by Reviewer X5Dv, and the well-structured presentation indicates a substantial contribution to the field. The concerns raised by Reviewers nAvZ and PYdg, while important, do not detract from the paper's core value and are addressable in future iterations. These points indicate that the paper has met the threshold for higher recognition within the conference.

---

### Decision · Program_Chairs · 2024-01-16

Accept (spotlight)